# Characterization of Berry Pomace Powders as Dietary Fiber-Rich Food Ingredients with Functional Properties

**DOI:** 10.3390/foods11050716

**Published:** 2022-02-28

**Authors:** Ieva Jurevičiūtė, Milda Keršienė, Loreta Bašinskienė, Daiva Leskauskaitė, Ina Jasutienė

**Affiliations:** Department of Food Science and Technology, Kaunas University of Technology, LT-50254 Kaunas, Lithuania; ieva.jureviciute@ktu.lt (I.J.); milda.kersiene@ktu.lt (M.K.); loreta.basinskiene@ktu.lt (L.B.); daiva.leskauskaite@ktu.lt (D.L.)

**Keywords:** berry pomace, dietary fiber, techno-functional properties, hypoglycemic, hypolipidemic properties

## Abstract

This study aimed to evaluate and compare the dried pomace powder of cranberries, lingonberries, sea buckthorns, and black currants as potential food ingredients with functional properties. The composition and several physicochemical and adsorption properties associated with their functionality were investigated. Tested berry pomace powders were rich in dietary soluble fiber (4.92–12.74 g/100 g DM) and insoluble fiber (40.95–65.36 g/100 g DM). The highest level of total phenolics was observed in the black currant pomace (11.09 GAE/g DM), whereas the sea buckthorn pomace revealed the highest protein concentration (21.09 g/100 g DM). All the berry pomace powders that were tested exhibited good water-holding capacity (2.78–4.24 g/g) and swelling capacity (4.99–9.98 mL/g), and poor oil-binding capacity (1.09–1.57 g/g). The strongest hypoglycemic properties were observed for the lingonberry and black currant pomace powders. The berry pomace powders presented effective in vitro hypolipidemic properties. The cholesterol-binding capacities ranged from 21.11 to 23.13 mg/g. The black currant and cranberry pomace powders demonstrated higher sodium-cholate-binding capacity than those of the lingonberry and sea buckthorn pomace powders. This study shows promising results that the powders of tested berry pomace could be used for further application in foods.

## 1. Introduction

The processing of berry juice, wine, or other beverages results in a considerable amount of pomace, including skins, seeds, and, occasionally, stalks. Pomace has been estimated at 30% of the total grape use in winemaking [1] or 60% of the total cranberry use in juice production [2]. However, berry pomace is no longer seen as a by-product and is further processed as a value-added food ingredient [3]. Multiple studies reported that antioxidant flavone fractions were extracted from berry pomace with high efficiency and evidenced strong free-radical-scavenging capacities [4,5,6]. Due to a high carbohydrate fraction, berry pomace can be considered a source of dietary fiber (DF).

The DF content in the pomace and the ratio between the soluble and insoluble fiber depend on the source and processing conditions during fiber isolation [7]. The pomace is typically deep-frozen at a minimum of −20 °C [8], milled, and dried at a temperature greater than 60 °C, resulting in a dry powder product. Reißner et al. [9] analyzed the chemical composition of dried black currant, red currant, gooseberry, rowanberry, and chokeberry pomaces. The authors reported that the insoluble fiber content in the dried pomaces varied between approximately 50.0 and 60.0 g/100 g DM and that the soluble fiber content varied between approximately 4.0 and 7.0 g/100 g DM. Other studies have highlighted the excellent technological properties (viscosity enhancement, swelling, water- and oil-holding capacities) of berry pomace [10]. Pomace from wine grapes [11] and pineapples [12] showed promising potential for use as an ingredient in the production of meat and dairy products to improve their stability and texture.

Moreover, the DF-rich berry pomace can promote health benefits, such as reducing preprandial cholesterol and postprandial blood glucose levels [13], enhancing gastrointestinal immunity [14], and increasing the satiety of consumers [15]. There are several pathways described regarding how DF influences glucose and cholesterol levels. For example, it increases the viscosity of the contents of the small intestine and retards the diffusion of glucose and lipids; it adsorbs glucose, cholesterol, and bile salts and decreases the levels of available glucose and lipids; it inhibits the activity of α-amylase and α-glucosidase; and it increases the formation of short-chain fatty acids in the gut and inhibits the formation of cholesterol [16,17].

Although the beneficial effects of DF have been well documented, their ability to deliver or promote these properties is highly dependent on their intake, as well as the source, physicochemical properties, and chemical composition. Thus, the characterization of berry pomace as a source of DF requires substantial knowledge regarding hypoglycemic activity and hypolipidemic effects. These properties of the berry pomace are particularly important when it is used as a food ingredient and the potential physiological effects must be predicted.

Therefore, the objective of this study was to analyze the dried coproducts from cranberry, lingonberry, sea buckthorn, and black currant juice extraction (pomace) as sources of DF. The study was designed to investigate the chemical composition, physicochemical properties (swelling capacity, water-holding capacity, oil-holding capacity), and functional properties (glucose adsorption capacity, sodium cholate, and cholesterol-binding capacity) of the dried berry pomaces. This study provides useful insight on the similarities and differences between four berry varieties and will aid in the selection of suitable dietary fiber-rich berry pomaces as functional ingredients for further application in foods.

## 2. Materials and Methods

### 2.1. Materials

Four berry varieties were used to produce berry pomace powder (PP): cranberries, lingonberries, sea buckthorns, and black currants. The berry pomaces were obtained using frozen berries that were donated by the Fudo Company (Kaunas, Lithuania). The berries were thawed and pressed in a Philips HR1880/01 juicer. The resulting pomace was immediately dried in a hot air dryer at 35 °C (to achieve a moisture content below 6%) and ground in a high-speed centrifugal mill (Retsch ZM200, Haan, Germany) using a 0.5-mm sieve. The resulting berry pomace powders were stored in sealed glass pots and refrigerated at 4 °C. 

### 2.2. Proximate Chemical Composition Analysis

The proximate chemical composition of the berry PP was determined using standard AOAC methods [18]. The moisture content was determined by drying a 3 g sample at 105 °C to constant mass. The protein content was analyzed according to the Kjeldahl procedure (conversion factor of nitrogen to crude protein—5.3). The fat content was determined by acid hydrolysis and subsequent extraction using petroleum ether in a Soxhlet apparatus. Ashing was performed on a 2 to 3 g sample after incineration in a muffle furnace at 550 °C for 4 h. The soluble dietary fiber (SDF) and insoluble dietary fiber (IDF) were determined according to the enzymatic-gravimetric method using a total dietary fiber kit (Megazyme, Ireland) based on AOAC 991.43 [19]. The remaining difference to 100% was considered to be comprised of non-DF carbohydrates. The IDF components, cellulose and acid-insoluble lignin, were identified using the Soest and Wine method [20], which was modified by McQueen and Nicholson [21].

The total phenol content in the extracts was determined using the Folin–Ciocalteu reagent according to the method described by Slinkard and Singleton [22]. The berry PP extracts were obtained by homogenizing 1 g of each sample with 10 mL of acidified methanol (0.1% hydrochloric acid), followed by vortexing for 1 min. The samples were then allowed to stand overnight in the dark at 4 °C. Next, the samples were centrifuged at 5000 rpm for 10 min at 4 °C, and the supernatants were collected and centrifuged again at 5000 rpm for 10 min at 4 °C. The supernatants were analyzed for total phenolic content (TPC) according Smolskaite et al. [23]. 

### 2.3. Determination of the Techno-Functional Properties

#### 2.3.1. Hydration Properties

The water-holding capacity (WHC) and swelling capacity (SC) of the berry PP were determined according to the procedures described by Luo et al. [24]. To determine the WHC, 0.5 g of berry PP was mixed with 10 mL of deionized water (at pH 2 and pH 7) and hydrated at room temperature for 24 h. The samples were centrifuged at 3000× *g* for 20 min (MPW-260R, MPW Med. Instruments, Poland) and the supernatants were carefully removed. The weight of the residue after hydration was recorded and the WHC was calculated as follows: (1)$$\mathrm{WHC}\;\left(g/g\right)=\frac{m_{2}-m_{1}}{m_{1}}$$
where m_1_ is the dry sample mass (g) before hydration and m_2_ is the sample mass (g) after hydration.

To determine the SC, 0.2 g of each berry pomace was weighed in a measuring tube that had 0.1 mL gradations, and the initial volume (mL) of the samples was measured. Subsequently, the samples were mixed with 5 mL of distilled water and maintained at room temperature for 24 h. The volumes (mL) of the hydrated samples were recorded and the SC was calculated as follows: (2)$$\mathrm{SC}\;\left(\mathrm{mL}/g\right)=\frac{\left(V_{2}-V_{1}\right)}{m}$$
where V_1_ is the sample volume before hydration, V_2_ is the volume after hydration, and m is the initial weight of the sample (g).

#### 2.3.2. Oil-Holding Capacity

To determine the oil-holding capacity (OHC), 1 g of berry PP was mixed with 20 mL of sunflower oil in a centrifuge tube and incubated at 37 °C for 1 h [24]. After incubation, the samples were centrifuged at 3000× *g* for 15 min (MPW-260R, MPW Med. Instruments, Poland) and the supernatants were carefully removed. The weight of the berry pomaces was recorded after incubation with the oil, and the OHC was calculated as follows: (3)$$\mathrm{OBC}\;\left(g/g\right)=\frac{m_{2}-m_{1}}{m_{1}}$$
where m_1_ is the weight of the sample (g) before incubation with the oil and m_2_ is the weight of the sample (g) after incubation with the oil.

### 2.4. Functional Properties

#### 2.4.1. Glucose Adsorption Capacity

The glucose adsorption capacity (GAC) of the berry PP was determined according to the method described by Bhutkar et al. [25], with certain modifications. The berry PP (1 g) was mixed with 25 mL of glucose solution of varying concentrations (0, 5, 10, 50, and 100 mmol/L) and incubated for 6 h at 37 °C, with periodical mixing at 120 rpm (GFL 1092, Thermolab, Germany). Next, the glucose concentration was determined using a glucose oxidase/peroxidase (GOPOD) assay kit (Megazyme, Ireland). The GAC was calculated as follows: (4)$$\mathrm{GAC}\;\left(\mathrm{mmol}/L\right)=\frac{C_{1-}\left(C_{2}-{\;C}_{3\;}\right)}{m}\times V$$
where C_1_ is the concentration of the original glucose solution; C_2_ is the concentration of the glucose in the samples with the berry PP and various concentrations of glucose solution (5, 10, 50, and 100 mmol/L) after 6 h of incubation; C_3_ is the concentration of glucose in the samples with the berry PP aqueous solution after 6 h of incubation; m is the weight of the berry pomace (g); and V is the volume of the sample (mL).

#### 2.4.2. Glucose Diffusion 

The glucose diffusion was determined according to the method described by Bhutkar et al. [25], with certain modifications. In total, 1 g of berry PP was mixed with 25 mL of 20 mmol/L glucose solution or 25 mL of water. The mixtures were dialyzed against 200 mL of distilled water in dialysis bags at 37 °C, with periodical mixing at 120 rpm (GFL 1092, Thermolab, Germany). Samples from the dialysate were collected at various time intervals (30, 60, 120, and 180 min) and the glucose concentration was determined using a GOPOD assay kit (Megazyme, Ireland). A control sample was prepared that did not contain berry PP. The glucose dialysis retardation index (GDRI) was calculated as follows: (5)$$\mathrm{GDRI}\left(\%\right)=100-\frac{C_{2}-{\;C}_{3}}{C_{1\;}}\times 100$$
where C_1_ is the glucose content of the control, C_2_ is the glucose content of the samples with the berry pomace in the glucose solution, and C_3_ is the glucose content of the samples with the berry pomace in the aqueous solution.

#### 2.4.3. Cholesterol-Binding Capacity

The ability of the berry PP to bind cholesterol was determined using a modification of the method described by Zhang et al. [26], where egg yolk was used as a model system. In total, 1 g of the berry PP was mixed with 50 mL of fresh egg yolk that was previously diluted with 9 volumes of deionized water. A sample that did not contain the berry PP was used as a control. After the pH was adjusted to pH 2 and pH 7, the mixture was incubated in a water bath (GFL 1092, Thermolab, Hanover, Germany) for 2 h at 37 °C, with periodical mixing at 120 rpm. To remove the color, 10 mL of the mixture was mixed with 0.1 g of polyvinylpolypyrrolidone, stirred for 1 min, and filtered using a paper filter (QLDF-125-100, PRAT DUMAS, Couze-et-Saint-Front, France). After the decolorization, 2 mL of each sample was mixed with 8 mL of 96% ethanol to precipitate impurities, and the samples were centrifuged at 4000× *g* for 20 min (Labofuge 200, Thermo Scientific, Altrincham, UK). The ethanol was removed from the supernatants using a rotary evaporator (RV 10, IKA, Königswinter, Germany). The cholesterol concentration in the samples was determined by the spectrophotometric method described by Park [27]. The cholesterol concentration was determined against the standard curve generated from the standard cholesterol solution.

The cholesterol-binding capacity (CBC) was calculated as follows:(6)$$\mathrm{CBC}\;\left(\mathrm{mg}/g\right)=\frac{C1-C_{2}}{m}\times V$$
where C_1_ is the cholesterol concentration (mg/mL) in the control sample; C_2_ is the cholesterol concentration (mg/mL) in the sample with the berry pomace; m is the weight of the berry pomace (g); and V is the volume of the sample (mL).

#### 2.4.4. Sodium-Cholate-Binding Capacity

The sodium-cholate-binding capacity (NaChBC) of the berry PP was determined according to the method described by Xu et al. [28]. A total of 0.2 g of berry PP and 0.2 g of sodium cholate were mixed with 100 mL NaCl solution (125 mmol/L, pH 7) in a 250 mL flask. A control sample was prepared that did not contain berry PP. The samples were incubated at 37 °C for 2 h, with periodical mixing at 120 rpm in a water bath (GFL 1092, Thermolab, Germany). The samples were then centrifuged at 4000× *g* for 20 min (Labofuge 200, Thermo Scientific, UK). The amount of unbound sodium cholate in the supernatant was determined using the method described by Shen et al. [29]. Briefly, 0.5 mL of supernatant was mixed with 4.5 mL of 42% *w*/*w* sulfuric acid for 20 min at 70 °C. The concentration of sodium cholate was determined from the calibration curve by measuring the absorbance of the solution at 387 nm (Thermo Scientific, UK). The NaChBC of the berry pomace was calculated as follows:(7)$$\mathrm{NaChBC}\;\left(\mathrm{mg}/g\right)\;=\;\frac{C_{1}-C_{2}}{m}\times V$$
where C_1_ is the concentration of sodium cholate in the control sample; C_2_ is the concentration of sodium cholate in the sample with the berry pomace; m is the weight of the berry pomace (g); and V is the volume of the sample (mL).

### 2.5. Statistical Analysis

All experiments were conducted in triplicate; the data are presented as the average value ± the standard deviation. The significant differences observed between the samples were evaluated using a *t*-test comparison, assuming a significant statistical difference when the *p* value was less than 0.05. The data were analyzed using Graph Pad Prism software.

## 3. Results and Discussion

### 3.1. Chemical Composition and Functional Properties of the Berry PP

The proximate composition of the cranberry, lingonberry, sea buckthorn, and black currant PP is presented in Table 1. The PP of all four types of berries were composed mainly of dietary fiber, with smaller amounts of protein, fat, and ash. The protein content in the berry PP varied widely, from 21.09 ± 0.36 g/100 g DM for sea buckthorn PP to 7.4 ± 0.06 g/100 g DM for cranberry PP. The measured fat content was higher in the black currant PP (13.85 ± 0.27 g/100 g DM). The cranberry, lingonberry, and sea buckthorn PP contained 9.83 ± 0.46, 12.68 ± 0.39, and 12.95 ± 0.44 g/100 g DM of fat, respectively. The total dietary fiber (TDF) content ranged from 49.24 ± 0.95 g/100 g DM in the black currant PP to 73.85 ± 0.83 g/100 g DM in the lingonberry PP. The results of the analyses demonstrated the predominance of IDF in all the berry PP. Most of the IDF consisted of cellulose and acid-insoluble lignin. The cellulose content ranged from 7.45 ± 1.04 g/100 g DM in the black currant PP to 17.89 ± 1.88 g/100 g DM in the lingonberry PP. The acid-insoluble lignin content was approximately 39 g/100 g DM for the sea buckthorn and cranberry PP, 42 g/100 g DM for the lingonberry PP, and 30 g/100 g DM for the black currant PP. A higher ratio between the SDF and IDF was determined for the cranberry and black currant PP (0.21 and 0.20, respectively) and a lower ratio was observed for the lingonberry and sea buckthorn PP (0.13 and 0.08, respectively). 

Data on the chemical composition of berry pomace in the literature are scarce, and the reported values are not comparable due to the differences in pomace preparation and the analytical methods used. However, in the case of cranberry pomace, certain general and comparable compositional features have been established. Andreani and Karboune [30] reported a similar protein (7.6 g/100 g DM), fat (6.3 g/100 g DM), and ash (0.9 g/100 g DM) content and a slightly lower value for the total dietary fiber (63 g/100 g DM) for dried cranberry pomace. Data in the literature on the basic components of dried black currant pomace vary widely, for example, from 11.8 to 15.7 g/100 g DM of protein, 2.5 to 20.2 g/100 g DM of fat, 2.3 to 4.1 g/100 g DM of ash, and 59.1 to 67.4 g/100 g DM of total dietary fiber [9,31,32]. This high variability may be due to differences in the cultivation, ripeness, and processing conditions of the berry pomace. Regarding the chemical composition of the sea buckthorn pomace, obtained results for the protein and ash content were similar to those reported by Ben-Mahmoud et al. [33] for dried sea buckthorn pomace (20.87 and 2.02%, respectively). The fat content observed in the present study was similar to that reported in studies performed by Ben-Mahmoud et al. [33] and Nour et al. [34]. The authors reported that the sea buckthorn pomace that was used as a supplement in animal feed consisted of 10.52 to 20.05% fat. However, the data in the literature on fiber content only include the crude fiber content in dried sea buckthorn pomace, which ranges from 18.3 to 19.86% [33,34]. According to Hao et al. [35], sea buckthorn pomace contains 29.27 to 32.11% DM of neutral detergent fiber and 20.45 to 22.9% DM of acid detergent fiber.

Most of the berry sugars and phenolic compounds were extracted into the juice. However, the residual sugar content in the berry PP was characterized by a great variability and ranged from 1.77 g/100 g DM for sea buckthorn PP to 24.04 g/100 g DM for black currant PP. The total phenolic content in the PP was also dependent on the type of berry and varied from 3.89 to 11.06 GAE/g DM. This content was higher than the total phenol content observed in seedless black currant pomace [32] and was similar to the results reported for blueberry and cranberry pomaces [36]. The fact that black currant PP contained the highest quantity of sugars and phenolic compounds indicates that the effectiveness of the juice extraction from the black currants was low.

The chemical composition of berry PP, particularly the amount and proportion of the SDF and IDF, leads to great variation in their functionality (Table 2).

The physicochemical properties of the berry PP, such as the WHC, SC, and OBC, are important from a technological and physiological standpoint. They are essential for the successful introduction of fiber-rich PP as a food ingredient and contribute to the textural properties and stability of formulated foods [3]. Additionally, the physicochemical properties of fiber-rich food ingredients are relevant for their hypoglycemic activity and hypolipidemic effects [37].

The WHC and SC of the berry PP at pH 7 ranged from 2.78 to 4.24 g/g and 4.99 to 9.98 mL/g, respectively. The highest WHC and SC were recorded for sea buckthorn PP. Similar values were observed for the WHC and SC when measured at pH 2. Previous studies on the hydration properties of red raspberry pomace reported similar results (WHC of 2.62 g/g and SC of 6.93 mL/g) [38]. A study conducted on pear pomace also exhibited similar WHC and SC values (3.44 g/g and 5.09 mL/g, respectively) [39]. Gouw et al. [36] reported higher WHC and lower SC values for dried berry pomace (a WHC of 8.70 g/g for cranberry pomace and 7.71 g/g for red raspberry pomace, and an SC of 5.87 mL/g for cranberry pomace and 2.88 mL/g for red raspberry pomace). The OHC of the berry PP in the present study ranged from 1.09 to 1.57 g/g and was highest for the cranberry PP, while the sea buckthorn PP showed the lowest value regarding the OHC. A high SDF content in cranberry PP may contribute to an increase in the OHC [40]. Previous studies reported higher OHC values for red raspberry pomace (2.44 g/g, [38]), cranberry and blueberry pomace (1.97 and 1.96 g/g, respectively, [36]), and pear pomace (1.82 g/g, [39]) and considerably higher OHC values for orange and grapefruit DF (3.62 and 8.20 g/g, respectively, [41]).

When berry PP is observed to have good hydration properties, the potential for its use as a food ingredient with a certain functionality can be predicted. This functionality may be reflected in certain physiological effects, such as hypoglycemic and hypolipidemic effects.

### 3.2. In Vitro Hypoglycemic Effects

The GAC is an index that characterizes the behavior of fiber in glucose adsorption during gastrointestinal transit in vitro [42]. The GAC is associated with the physical properties of the network structure of DF, which may cause the adsorption or inclusion of small sugar molecules. Therefore, the consumption of DF leads to the inhibition of postprandial hyperglycemia [41]. The GAC of the berry PP tested in the present study is presented in Figure 1. The GAC was dependent on the molar concentration of glucose. At low glucose concentrations (5 and 10 mmol/L), all the berry PP exhibited a small and similar capacity to adsorb glucose, with values ranging from 0.05 to 0.12 mmol/g. As the concentration of glucose increased to 100 mmol/L, the berry PP effectively increased the GAC, with the black currant and sea buckthorn PP showing the highest values (1.20 and 1.31 mmol/g, respectively). These results indicated that the berry PP were effective at adsorbing glucose only at high concentrations (50 and 100 mmol/L), although the quantity of glucose bound to the berry PP was dependent on the source of the PP. Pomaces from various fruit and vegetable sources have been reported to exhibit a GAC, most likely due to a high IDF content. For example, the GAC of DF extracted from carrots was shown to be dependent on the extraction method used and ranged from 2.43 to 2.63 mmol/g after incubation in a 100 mmol/L glucose solution [43]. In a separate report, de-oiled red raspberry pomace and DF extracted from de-oiled red raspberry pomace showed GAC values of 3.42 mg/g and 1.73 mg/g, respectively [38]. Huang et al. [44] observed a GAC value of 0.89 mmol/g for psyllium pomace after incubation in a 200 mmol/L glucose solution. The authors also demonstrated that, with increased glucose concentrations, higher amounts of glucose were bound. The results of the present study suggest that the berry PP can maintain a low level of glucose in the small intestine, which reduces the contact with the intestinal tract and, consequently, decreases the potential for postprandial hyperglycemia.

The GDRI is an index that is used to characterize the hypoglycemic effects of DF. The GDRI was created to predict the effect of fiber on delaying glucose absorption in the gastrointestinal tract due to glucose entrapment by the network formed by DF [45]. The in vitro glucose diffusion and GDRI of the berry PP tested in the present study are presented in Table 3. 

During a time interval from 0 to 180 min, the glucose concentration increased in the dialysate for all the samples. In the case of the sea buckthorn PP, the registered increase in released glucose ranged from 1.00 mmol (after 30 min) to 2.05 mmol (after 180 min). For the lingonberry PP, the glucose concentration in the dialysate increased from 0.83 mmol (after 30 min) to 1.83 mmol (after 180 min). However, the control sample (that did not contain the berry PP) displayed a significantly higher glucose release (*p* < 0.05) at various time intervals compared to the berry PP samples. By calculating the GDRI, which indicates the effect of the berry PP on the delay in glucose absorption in the gastrointestinal tract, it was obvious that all the berry PP decreased the amount of glucose dialyzed regarding the control sample. After 30 min, the lingonberry PP exhibited the highest GDRI (27.49%), followed by the black currant PP (16.61%), the sea buckthorn PP (13.30%), and the cranberry PP (10.40%). Similar values were observed after 60, 130, and 180 min of dialysis. Acquired findings are in accordance with the observations reported for DF isolated from orange and psyllium pomace [44] and DF-rich coffee parchment [46]. However, the values observed in the present study were lower than those reported for other sources of DF, such as oats, barley, and psyllium husk [47]. This capability of DF to delay glucose absorption is associated with the increased viscosity of soluble polysaccharides and the network formed by the insoluble fiber that maintains the entrapped glucose [48]. In the present study, we assumed that the latter prevailed regarding the decrease in glucose absorption. According to our results, the lingonberry PP, which showed the highest amount of IDF and a moderate amount of SDF (65.36 and 8.49%, respectively), exhibited the highest GDRI values. Moreover, the cranberry PP, which showed a lower IDF content and a high SDF content (59.93 and 12.74%, respectively), exhibited the lowest GDRI.

### 3.3. In Vitro Hypolipidemic Effects

The CBC and NaChBC are valuable indicators for characterizing the hypolipidemic properties of fiber-rich berry PP [24]. The results recorded in the present study regarding these capacities are presented in Table 4. 

The CBC of the cranberry, lingonberry, sea buckthorn, and black currant PP at pH 7 were 23.13 ± 0.47 mg/g, 22.61 ± 0.45 mg/g, 22.75 ± 0.46 mg/g, and 21.11 ± 0.42 mg/g, respectively. These results were in line with the OBC of tested berry PP (Table 4)—the highest OBC was observed for the cranberry PP and the lowest OBC was observed for the sea buckthorn and black currant PP. At pH 2, all the berry PP tested bind a lower amount of cholesterol. These results corroborate those obtained by other researchers who observed that cholesterol adsorption by DF occurred mainly in the intestines rather than the stomach [49]. The cranberry PP demonstrated the highest CBC (21.91 ± 0.02 mg/g) and the lingonberry PP showed the lowest CBC at pH 2. A decrease in the cholesterol levels in the small intestine is associated with SDF rather than IDF [50]. In the present study, the cranberry PP contained the highest amount of SDF (12.74 ± 0.09 mg/100 g DM), which may be associated with the improved CBC observed for this PP compared with the others. There is an abundance of scientific evidence regarding the antihyperlipidemic activity of polyphenol-rich berries on the key steps of lipid digestion and absorption [37]. However, data on the capacity of the DF from berries to bind cholesterol are not so abundant. In a previous study, the cholesterol adsorption capacity and the total DF for de-oiled red raspberry pomace were reported at 6.04 mg/g and 2.61 mg/g (at pH 7), respectively [38]. Additionally, data exists on the hypolipidemic effects of DF from various fruits. Dried fiber-rich orange pomace contained 17.31% of SDF and showed 6.89 mg/g DM of CBC [44]. In the case of dried peach pomace that contained 10.0 g/100 g of SDF, the recorded CBC was 3.88 ± 0.48 mg/g at pH 7 and 1.27 ± 0.02 mg/g at pH 2 [39]. Wang et al. [41] reported that the CBC of SDF from various citrus fruit peels at pH 2 ranged from 1.66 ± 0.29 mg/g to 10.88 ± 0.32 mg/g, while the values were recorded at 3.22 ± 0.38 mg/g to 17.90 ± 0.17 mg/g at pH 7. Moreover, the CBC of SDF from wheat bran was considerably lower at 2.17 ± 0.07 mg/g and 3.48 ± 0.03 mg/g at pH 2 and pH 7, respectively [26].

DF is also capable of trapping bile acids in the small intestine. In the human body, more than 90% of bile acids exist in the form of binding substances, such as sodium cholate. Obtained results demonstrated that the binding capacity of sodium cholate for four DF-rich berry PP were different. The highest NaChBC was observed for the black currant PP (74.78 ± 1.39 mg/g) at pH 7, followed by the cranberry, lingonberry, and sea buckthorn PP. These bile-acid-binding values are comparable to those reported for modified millet bran DF [51] and are higher than those reported for DF in soybean-seed hulls, wheat bran, and apple peels [26], but they are considerably lower compared to the SDF recorded in citrus fruits peels [41].

Various mechanisms have been described regarding the reduction of the NaChBC by fiber. The main effects are associated with the viscous and gel-forming properties of soluble fiber and the presence of phenolic compounds. The presence of two types of binding sites on the bile salt micelles makes them suitable for binding both hydrophobic and hydrophilic bioactive molecules [52]. Zhou and Wang [53] revealed that phenolic compounds may bind with sodium cholate monomers, dimers, and primary and secondary micelles mainly through hydrophobic interactions. In the present study, the black currant PP was high in phenolic compounds (11.06 ± 0.40 GAE/g DM), and this can impact the high capacity of this PP to bind sodium cholate. The cranberry PP also showed a high sodium-cholate-binding capacity and contained a higher amount of SDF (12.74 ± 0.09 mg/100 g DM) compared to the other berry PP. According to Ma et al. [54], the effectiveness of DF in increasing sodium cholate binding is mainly attributed to a high water SC and SDF content, which can increase the viscosity of the medium.

Thus, the berry PP can effectively absorb bile acids and other lipid substances in the small intestine and help to remove these substances while performing as a lipid scavenger [55]. A decrease in sodium cholate content invokes the conversion of cholesterol into sodium cholate for its supplementation, thereby promoting the consumption of cholesterol [56]. Therefore, the adsorption capacity of DF for cholesterol and cholate is a good indicator for the adsorption of lipophilic substances. The differences in the hypolipidemic effects of the berry PP tested in the present study were associated with their composition.

## 4. Conclusions

The powders obtained from the cranberry, lingonberry, sea buckthorn, and black currant pomaces analyzed in the present study demonstrated physicochemical and functional properties that support their use as fiber-rich food ingredients. The drying and subsequent milling of all berry pomaces tested resulted in powders that contained considerable amounts of total dietary fiber (49.24–73.85 g/100 g DM). The sea buckthorn PP was observed to contain the highest amount of protein (21.09 g/100 g DM). The lingonberry PP was different to the other varieties due to its high insoluble fiber content (65.36 g/100 g DM) and predominant amount of acid-insoluble lignin (42.08 g/100 g DM) and cellulose (17.89 g/100 g DM). The highest level of SDF was observed in the cranberry PP (12.74 g/100 g DM), while the black currant PP contained the highest amount of total phenolics (11.06 GAE/g DM). Knowledge of the chemical composition of berry PP is important for interpreting differences in the physicochemical and functional properties. All the berry pomace powders tested exhibited good hydration properties with the highest WHC and SC recorded for the sea buckthorn PP (4.24 g/g and 9.98 mL/g, respectively). The OBC of the berry PP ranged from 1.09 to 1.57 g/g and was highest for the cranberry PP, while the sea buckthorn PP showed the lowest value for the OBC. The functional potential was shown for all berry PP tested and was based on the findings that the in vitro hypoglycemic and hypolipidemic properties of the berry PP were similar and, in certain cases, better than other DF powders obtained from fruit and vegetable processing by-products. The highest GAC and GDRI were observed for the lingonberry and black currant PP, and these values were considerably lower for the cranberry PP. The CBC of berry PP ranged from 21.11 to 23.13 mg/g. The black currant and cranberry PP demonstrated a higher NaChBC than that of the lingonberry and sea buckthorn PP. The results of this study provide valuable information for the potential application of berry PP in foods as a functional ingredient. Further research that focuses on food models created using berry PP and their behavior during in vitro digestion is necessary to evaluate the functionality of berry PP as a dietary fiber-rich food ingredient. Further in vivo investigations on the functional properties of berry pomace are needed to confirm their health benefits.

## Figures and Tables

**Figure 1 foods-11-00716-f001:**
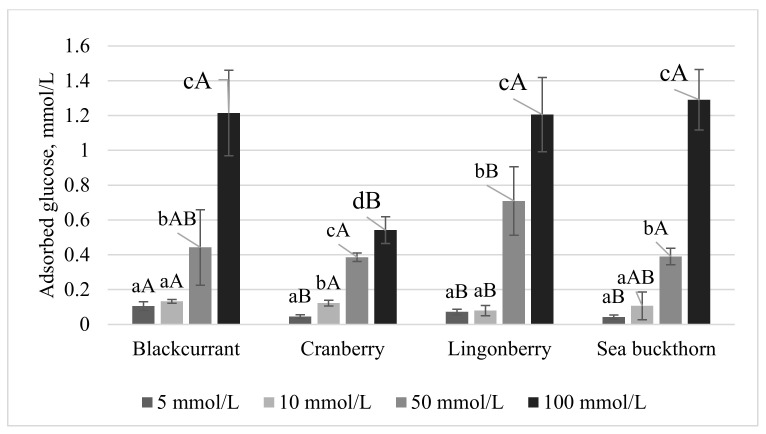
Glucose adsorption capacity at different concentrations of glucose (5, 10, 50, and 100 mmol/L) of berry PP. Values expressed as average ± st. dev. Different letters among columns indicate significant (*p* < 0.05) differences (lower case—among concentrations; capital letters—among berry PP with the same glucose concentration).

**Table 1 foods-11-00716-t001:** Proximate chemical composition of berry pomace powder, g/100 g DM.

Parameters	Cranberry	Lingonberry	Sea Buckthorn	Black Currant
Moisture	5.57 ^c^ ± 0.11	3.41 ^a^ ± 0.04	4.17 ^b^ ± 0.04	7.97 ^d^ ± 0.10
Ash	0.96 ^a^ ± 0.04	1.18 ^b^ ± 0.01	1.38 ^c^ ± 0.04	3.82 ^d^ ± 0.02
Protein (N × 6.25)	7.4 ^a^ ± 0.06	8.60 ^b^ ± 0.27	21.09 ^c^ ± 0.36	9.05 ^b^ ± 0.26
Fat	9.83 ^a^ ± 0.46	12.68 ^b^ ± 0.39	12.57 ^b^ ± 0.20	13.85 ^c^ ± 0.27
Total dietary fiber	72.67 ^c^ ± 1.55	73.85 ^c^ ± 0.83	63.61 ^b^ ± 1.64	49.24 ^a^ ± 0.95
Total insoluble dietary fiber:	59.93 ^b^ ± 1.46	65.36 ^c^ ± 0.67	58.69 ^b^ ± 0.96	40.95 ^a^ ± 0.78
Cellulose	17.14 ^a^ ± 2.01	17.89 ^a^ ± 1.88	13.62 ^b^ ± 2.24	7.45 ^c^ ± 1.04
Acid-insoluble lignin	39.58 ^a^ ± 1.55	42.08 ^b^ ± 1.73	39.23 ^a^ ± 2.09	30.16 ^c^ ± 1.91
Total soluble dietary fiber	12.74 ^c^ ± 0.09	8.49 ^b^ ± 0.05	4.92 ^a^ ± 0.68	8.29 ^b^ ± 0.17
Soluble DF/insoluble DF ratio	0.21 ^a^	0.13 ^b^	0.08 ^c^	0.20 ^a^
Carbohydrates *	9.14 ^a^	4.43 ^b^	1.35 ^c^	24.04 ^d^
Total phenolic content, GAE/g DM	3.89 ^a^ ± 0.29	6.26 ^b^ ± 0.23	5.73 ^c^ ± 0.02	11.06 ^d^ ± 0.40

* Calculated as 100 − (fat + ash + protein + total dietary fiber). Different letters indicate statistically significant differences in row, *p* < 0.05.

**Table 2 foods-11-00716-t002:** Physicochemical properties of berry pomace powder.

Powder of Pomace	Water Holding Capacity, (g/g)		Swelling Capacity, (mL/g)		Oil Binding Capacity, (g/g)
	pH 2	pH 7	pH 2	pH 7	
Black currant	2.78 ^a^ ± 0.04	2.78 ^a^ ± 0.02	4.14 ^a^ ± 0.25	4.99 ^a^ ± 0.21	1.14 ^a^ ± 0.01
Cranberry	3.83 ^b^ ± 0.09	3.87 ^b^ ± 0.18	7.95 ^b^ ± 0.40	7.99 ^b^ ± 0.40	1.57 ^b^ ± 0.05
Lingonberry	3.28 ^c^ ± 0.05	3.27 ^c^ ± 0.02	7.90 ^b^ ± 0.40	8.00 ^b^ ± 0.40	1.46 ^c^ ± 0.05
Sea buckthorn	4.32 ^d^ ± 0.04	4.24 ^d^ ± 0.15	10.95 ^c^ ± 0.50	9.98 ^c^ ± 0.55	1.09 ^d^ ± 0.02

Different letters indicate statistically significant differences in column, *p* < 0.05.

**Table 3 foods-11-00716-t003:** Effect of berry PP on glucose diffusion from 30 to 180 min of incubation at 37 °C.

Powder of Pomace	Glucose Concentration in the Dialysate, (mmol/g)			
	30 min	60 min	120 min	180 min
Control	1.15 ^aA^ ± 0.01	1.58 ^bA^ ± 0.02	1.98 ^cA^ ± 0.03	2.10 ^cA^ ± 0.1
Black currant	0.96 ^aB^ ± 0.05 (16.61) *	1.40 ^bB^ ± 0.07 (11.39)	1.77 ^cB^± 0.09 (10.96)	1.92 ^cB^ ± 0.10 (8.46)
Cranberry	1.03 ^aB^ ± 0.05 (10.40)	1.48 ^bB^ ± 0.07 (6.40)	1.86 ^cB^ ± 0.09 (5.97)	2.05 ^cA^ ± 0.10 (2.22)
Lingonberry	0.83 ^aC^ ± 0.04 (27.49)	1.17 ^bC^ ± 0.06 (25.74)	1.62 ^cC^ ± 0.08 (18.50)	1.83 ^cB^ ± 0.09 (12.76)
Sea buckthorn	1.00 ^aB^ ± 0.05 (13.30)	1.48 ^bB^ ± 0.07 (6.13)	1.93 ^cAB^ ± 0.10 (2.49)	2.05 ^cA^ ± 0.10 (2.23)

* Data in parentheses present the glucose dialysis retardation indexes of various samples. Different letters among columns indicate significant (*p* < 0.05) differences (lower case—among time of dialysis; capital letters—among berry PP at the same time of dialysis).

**Table 4 foods-11-00716-t004:** Cholesterol- and sodium-cholate-binding capacities of berry pomace powder.

Powder of Pomace	Cholesterol-Binding Capacity, mg/g		Sodium-Cholate-Binding Capacity, mg/g
	pH 2	pH 7	pH 7
Black currant	18.02 ^a^ ± 0.01	21.11 ^a^ ± 0.42	74.78 ^a^ ± 1.39
Cranberry	21.91 ^b^ ± 0.02	23.13 ^b^ ± 0.47	52.68 ^b^ ± 2.07
Lingonberry	14.16 ^c^ ± 0.01	22.61 ^b^ ± 0.45	40.71 ^c^ ± 2.78
Sea buckthorn	15.11 ^d^ ± 0.06	22.75 ^b^ ± 0.46	24.66 ^d^ ± 5.80

Different letters among columns indicate significant differences (*p* < 0.05).

## Data Availability

The study did not report any data.

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
