# Peer review of "Characterization of Berry Pomace Powders as Dietary Fiber-Rich Food Ingredients with Functional Properties"

_foods, 2022, doi:10.3390/foods11050716_

Round 1

Reviewer 1 Report

I have read carefully the paper entitled "Characterization of Berry Pomace Powders as Dietary Fiber-rich Food Ingredients with Functional Properties " and find it very insightful and significant for the scientific community. However, certain issues must be answered. In my view, the manuscript needs to be a minor revision.

My suggestions are listed as follows:

Abstract

Abstract has to have a little promising moment in the end. Please add why your results are good, further perspective(s), etc. I suggest reducing number of words, to reorganizing this section to be more attractive to read

  1. Introduction

The complete text needs to be in passive form, avoiding „we are, we do, we investigated, our results, our funding...“ – this suggestion is for the complete text, but particularly for the aims in the Introduction part.

  1. Material and methods

Before the first subsection, add an explanation of further steps, providing the concept of study. Optionally, you can add a graphical abstract.

In Eq. (5) add unit (%)

Add unit for Eq. (7)

  1. Results

Presenting the results of this study is good, but can be improved. You can really upgrade your investigation using Principal component analysis (PCA) for better visualization of the obtained results.

In Table 2, replace “.” to “,”, before units

  1. Conclusions

Add potential investigation in future, as well as advantages and disadvantages of this study.

Author Response

Responses to the comments presented in reviews to the manuscript:

Characterization of Berry Pomace Powders as Dietary Fiber-rich Food Ingredients with Functional Properties, by Ieva Jurevičiūtė, Milda Keršienė, Loreta Bašinskienė, Daiva Leskauskaitė and Ina Jasutienė

First of all, the authors would like to thank the Reviewers for their time and effort dedicated to reviewing the manuscript. We are sure that all valuable comments and suggestions will help us to improve the manuscript and also contribute to the effectiveness of our further work and publications.

Responses to the comments in the Review I

Comment:

Abstract has to have a little promising moment in the end. Please add why your results are good, further perspective(s), etc. I suggest reducing number of words, to reorganizing this section to be more attractive to read

Answer:

Thank you for your comment. We have been shortened the abstract and reorganized it.

Comment:

The complete text needs to be in passive form, avoiding „we are, we do, we investigated, our results, our funding...“ – this suggestion is for the complete text, but particularly for the aims in the Introduction part.

Answer:

Thank you for your comment. The text has been corrected according reviewer suggestions.

Comment:

Before the first subsection, add an explanation of further steps, providing the concept of study. Optionally, you can add a graphical abstract.

Answer:

We provided details of experiment design at the end of the previous section (Introduction). So, we avoid duplication of information by providing the concept of study in Materials and Methods section. The graphical abstract is added and we hope it would help to understand the design of study.

Comment:

In Eq. (5) add unit (%)

Add unit for Eq. (7)

Answer:

Thank you for your careful reading. Manuscript has been corrected.

Comment:

Presenting the results of this study is good, but can be improved. You can really upgrade your investigation using Principal component analysis (PCA) for better visualization of the obtained results.

Answer:

Thanks for suggestion to use PCA. We tried to use this method of visualization, but unfortunately Principal component analysis did not show clear separation of the samples. It was investigated four kinds of pomace and a lot of different parameters. May be a better result can be obtained if for PCA we hadn’t used all studied parameters but only selected of them. We will use such suggestion in the future investigations.

Comment:

In Table 2, replace “.” to “,”, before units

Answer:

Thank you for your comment. The text has been corrected.

Comment:

Conclusions. Add potential investigation in future, as well as advantages and disadvantages of this study.

Answer:

Authors agree with the comment of the Reviewer. The conclusions were revised and supplemented by: Further in vivo investigations on the functional properties of berry pomace are needed to confirm their health benefits.

Reviewer 2 Report

I reviewed the manuscript entitled Characterization of Berry Pomace Powders as Dietary Fiber rich Food Ingredients with Functional Properties. The manuscript is well written with background information, clearly stated objectives, and can grab readers attention. The conclusions are supported by the experiment and appropriate analysis. 

Line 109: what is the role of (or yogurt) here?

Line 140: replace 1 g of the berry PP with berry PP (1 g)

Equation 5: Please rewrite. The bracket is not correct

Table 1. statistical analysis must be conducted among pomace. Tables may have a footer…. What does it mean?

In TABLE 1. Total phenolic content, GRE/g DM ..In methodology, it says mg of gallic acid equivalents per g of dry weight sample (or yogurt) (mg 109 GAE/g dw)… What is GRE Here?

Table 2. replace Water holding capacity. g/g with Water holding capacity (g/g). Do the other changes too.

Table 2. Is statistical analysis comparison row or column wise? Please indicate

Table 2 footnote: Different letters indicate statistically significant differences in column, p< 0. .. p < 0.???? No value?

Table 3.  What is the star indicating in the first column related to Black currant?

In vitro should be Italics throughout the manuscript

References are not according to the journal. Please read the author's guidelines and adjust accordingly. 

Author Response

Responses to the comments presented in reviews to the manuscript:

Characterization of Berry Pomace Powders as Dietary Fiber-rich Food Ingredients with Functional Properties, by Ieva Jurevičiūtė, Milda Keršienė, Loreta Bašinskienė, Daiva Leskauskaitė and Ina Jasutienė

First of all, the authors would like to thank the Reviewers for their time and effort dedicated to reviewing the manuscript. We are sure that all valuable comments and suggestions will help us to improve the manuscript and also contribute to the effectiveness of our further work and publications.

Responses to the comments in the Review II

Thank you very much for careful reading. Authors agree with the comments of the Reviewer. Manuscript has been corrected and improved according comments.